# Improving the Drug Development Pipeline for Mycobacteria: Modelling Antibiotic Exposure in the Hollow Fibre Infection Model

**DOI:** 10.3390/antibiotics10121515

**Published:** 2021-12-10

**Authors:** Arundhati Maitra, Priya Solanki, Zahra Sadouki, Timothy D. McHugh, Frank Kloprogge

**Affiliations:** 1Institute for Global Health, University College London, London WC1N 1EH, UK; zahra.sadouki.15@ucl.ac.uk (Z.S.); f.kloprogge@ucl.ac.uk (F.K.); 2Centre for Clinical Microbiology, Royal Free Campus, University College London, Rowland Hill Street, London NW3 2PF, UK; p.solanki@ucl.ac.uk (P.S.); t.mchugh@ucl.ac.uk (T.D.M.)

**Keywords:** *Mycobacterium*, tuberculosis, hollow fibre, drug development

## Abstract

Mycobacterial infections are difficult to treat, requiring a combination of drugs and lengthy treatment times, thereby presenting a substantial burden to both the patient and health services worldwide. The limited treatment options available are under threat due to the emergence of antibiotic resistance in the pathogen, hence necessitating the development of new treatment regimens. Drug development processes are lengthy, resource intensive, and high-risk, which have contributed to market failure as demonstrated by pharmaceutical companies limiting their antimicrobial drug discovery programmes. Pre-clinical protocols evaluating treatment regimens that can mimic in vivo PK/PD attributes can underpin the drug development process. The hollow fibre infection model (HFIM) allows for the pathogen to be exposed to a single or a combination of agents at concentrations achieved in vivo–in plasma or at infection sites. Samples taken from the HFIM, depending on the analyses performed, provide information on the rate of bacterial killing and the emergence of resistance. Thereby, the HFIM is an effective means to investigate the efficacy of a drug combination. Although applicable to a wide variety of infections, the complexity of anti-mycobacterial drug discovery makes the information available from the HFIM invaluable as explored in this review.

## 1. Introduction

The genus *Mycobacterium* contains several important human and animal pathogens in addition to environmental species. *Mycobacterium tuberculosis* is the causative agent of tuberculosis (TB), a chronic infection primarily affecting the lungs. Even drug-sensitive (DS) TB infections are difficult to treat and require chemotherapy that involves a combination of antibiotics administered over a 6-month duration. This is due to the intrinsic resistance of the pathogen provided by its inherent characteristics such as an impermeable cell wall, active drug efflux systems, and the ability to switch between ‘actively growing’ and ‘metabolically dormant’ physiologies, making it less susceptible to attack by antimicrobial agents and host immune responses [1,2,3,4]. In addition, once internalised by macrophages, *M. tuberculosis* can escape lysosomal degradation and survive intracellularly creating a further barrier to drug exposure [5]. The host response sequesters the pathogen into clusters called granulomas—the hallmark structure of TB. Granulomas are a compact, organised aggregate of immune cells (mostly macrophages and lymphocytes) containing free bacilli and infected macrophages in the centre [6]. Thus, the site of the infection and the bacterium itself are difficult for many drugs to penetrate explaining, in part, the protracted treatment regimen. 

A dearth of new antimicrobial agents, reticence on behalf of pharma to limit these to mycobacteria, and the effectiveness of the standardised treatment for DS-TB has resulted in a therapy that has barely been modified over the past 40 years [7]. Although, the current standard therapy is successful in treating DS-TB, various factors can lead to treatment failure. These include but are not limited to: co-infections, infection by a resistant strain of *M. tuberculosis*, inappropriate patient management, and in some settings, inadequate or sub-standard antibiotics. The absence of options to treat patients who relapse or fail their treatment leads to the transmission of resistant infections within the community, further exacerbating the situation. Encouragingly over the past decade, new drugs and drug regimens have been trialled with considerable success in terms of treating resistant TB infections as well as shortening therapy times or both [8,9]. However, the armoury is limited and there is room for improvement in the drug development process for antimicrobials in general and antitubercular agents in particular. 

Non-tuberculous mycobacteria (NTM) constitute over 160 *Mycobacterium* species and are usually found in soil and water supplies. NTM are opportunistic pathogens commonly known to cause chronic, pulmonary infections, which like *M. tuberculosis*, are notoriously hard to treat [10]. The perceived prevalence of NTM infections is increasing globally, and in recent years there has been a heightened awareness of the clinical relevance of NTM infections [11]. Currently, treatment of NTM infections involves at least three drugs (including one macrolide: clarithromycin or azithromycin) over an 18–24-month period. However, despite the use of multiple drugs over a long period of time the treatment outcome remains poor with significant adverse effects [12]. Treatment of *M. abscessus* infections, for example, has a 25–58% cure rate emphasising the urgent need for novel drug regimens with shorter treatment durations [13].

Pre-clinical phases of antibacterial drug development include various in vitro assays to determine drug potency, evaluate toxicity, and identify drug targets. Microtitre plate formats combined with automated reagent dispensing systems have enabled the automation of assays and increased throughput for screening libraries of chemicals to identify ‘hit’ compounds. These assays are typically followed by the determination of pharmacokinetic/pharmacodynamic profiles of the ‘lead’ compounds to model drug efficacy and testing in animal models prior to their clinical evaluation. Microtitre plate assays are limited to testing conditions that remain static over the course of the assay and hence cannot reflect the changes in concentrations of a drug during treatment, and thereby do not provide the rate/success of bacterial elimination under dynamic conditions. On the other hand, animal models may not reflect human PK/PD values and it is not always possible to sample the same animal over several time points. In vitro, one-compartment PK/PD models consist of a diluent reservoir, an open central reservoir containing the organism of interest, and a waste reservoir. Drugs are added to the central reservoir and the elimination curve is modelled by adding diluent to the central reservoir at the same rate as media is eliminated from it. The disadvantages of the system include: (a) exposure of the worker to the pathological agent, (b) changes in bacterial number over time, (c) requirement for large volumes of drug and diluent, and (d) inability to model drugs with short half-lives. 

The hollow fibre infection model (HFIM) overcomes these shortcomings. It is a two-compartment bioreactor containing tubular hollow fibres made of semi-permeable membrane that are sealed at each end. The molecular weight cut off of the pores of the fibres is small, containing bacteria within the extra-capillary space (ECS) while allowing metabolites, drugs, and other small molecules to pass freely across [14,15,16]. The pathogen is contained in a small volume (typically <20 mL) ensuring worker safety while reaching population densities achieved in in vivo infections. Multiple samples can be extracted longitudinally to study the killing of the pathogen as well as emergence of resistance without significantly disturbing the bacterial population. Human PK/PD values can be precisely replicated as drugs equilibrate rapidly across the membrane and a combination of drugs can be tested simultaneously [17,18].

Hollow fibre cartridges were first used in the 1980’s for antibacterial testing and have since found widespread use [19,20]. The model has a 94% predictive accuracy for clinical therapeutic events, such as dose, concentrations/exposures optimal in patients, and expected rates of clinical efficacy [21,22]. The European Medicines Agency (EMA) supported the use of HFIM for the selection and development of antituberculosis drug regimens in 2015 [23,24]. The application of hollow fibre models to identify anti-mycobacterial compounds, select their pharmacodynamic target, support PK/PD analyses, and identify dose selection both in single drug and combination regimen studies will be discussed further on in this review. 

## 2. HFIM in Anti-Mycobacterial Drug Discovery—The Working Model

The hollow fibre cartridge is comprised of the ECS and the intra capillary spaces (ICS). The ICS is continuous with the diluent, central, and waste reservoirs (Figure 1). The molecular weight cut-off on the fibres allows for continuous replenishment of media and the addition/elimination of drugs to and from the ECS, while containing the movement of cells. The ECS can be inoculated with free bacilli (to test against extra-cellular pathogens) or macrophages infected with mycobacteria (to test against intra-cellular pathogens). The media components can be altered to replicate different physiological states of *M. tuberculosis* such as actively-replicating, metabolically dormant, and/or other environmental stress conditions. This was demonstrated by Louie et al. while studying the efficacy of moxifloxacin against *M. tuberculosis* in acidic environments as well as in a non-replicating persister phenotype [25]. 

Single or multiple drugs can be administered, either as a bolus or continuous infusion with the help of computerised syringe drivers. Elimination half-lives are typically mimicked by making use of peristaltic pumps, and antimicrobial combinations with substantially different elimination half-lives can be simulated through either first-order or zero-order infusion topping up, with the shorter half-life drug being the backbone. With the help of R-shiny web applications (https://pkpdia.shinyapps.io/hfs_app/, accessed on 9 December 2021, https://varacli.shinyapps.io/hollow_fiber_app/, accessed on 9 December 2021) users can convert in vivo pharmacokinetic profiles into in vitro pump settings in the HFIM [26,27].

The response of the pathogen to treatment is determined by an estimation of the bacterial load over the course of the experiment. At present, bacterial load during the treatment of *M. tuberculosis* in patients can be quantified using microscopy, serial sputum colony counting (SSCC), liquid culture using Mycobacterium Growth Indicator Tubes (MGIT)/time to positivity (TTP), and/or the Mycobacterium Bacterial Load Assay (MBLA) [28]. Many of these methods or their equivalents are used to enumerate bacterial load during the course of an HFIM experiment (Figure 2).

Using Ziehl–Neelsen staining methods followed by light microscopy as a quantification method has the disadvantage of having a low sensitivity limit (10^4^ CFU/mL) and cannot differentiate between live and dead bacilli. Staining with fluorescent dyes (e.g., SYTOX™ Green, calcein violet, propidium iodide, Nile red, and resazurin) followed by cell sorting allows for more precise measurement as well as differentiation of cells as live, injured, or dead adding information on bacterial health as well as load [29,30]. Measuring CFU is commonly used to quantify changes in bacterial load. However, heterogenous populations of *M. tuberculosis* cells include viable but nonculturable cells (VBNCs) that do not grow on solid media and growth in liquid media is considered to be more sensitive [31,32,33]. Time to positivity (TTP) readouts provided by the BD BACTEC™ MGIT™ systems can offer a semi-automated alternative to conventional CFU methods. However, both solid and liquid culture are time consuming and often do not give a precise total number of cells (in all physiological states) present in a given sample. The MBLA is a molecular assay that can be used to monitor bacterial load and has a sensitivity of 100 CFU/mL. The principle of the assay is based on the detection of *M. tuberculosis* 16S ribosomal RNA (rRNA), with an internal control (IC) using reverse transcription quantitative PCR (RT-qPCR) [28]. 

The other crucial information that the HFIM can provide is the emergence of selective-, cross-, or pan-resistance in the pathogen to drug treatment. Unlike in animal experiments, repeated sampling from the hollow fibre cartridge of the bacterial culture can be performed. Plating these samples on to media containing high concentrations of drugs can provide information on whether the given treatment enhances or suppresses the emergence of resistance. Combined with whole-genome sequencing technologies, the mechanisms underlying the evolution of resistance can also be studied. 

## 3. Application of HFIM in Detecting Novel or Repurposed Anti-Mycobacterial Compounds

The call to repurpose drugs used for other infections/indications, for the treatment of TB has been gaining momentum over the last decade [34,35]. Deshpande et al. initiated a programme to identify pre-existing pharmacophores with anti-mycobacterial activity and identified ceftazidime-avibactam and benzylpenicillin as the first hits followed by minocycline [36,37,38,39]. Deshpande et al. reported ceftazidime-avibactam to have a sterilising effect at par with standard TB therapy [36]. They tested ceftazidime-avibactam against rapidly growing, intracellular, and semidormant *M. tuberculosis* in the HFIM and recommended a dose of 100 mg/kg three times a day for children and up to 12 g in adults, a dose which is tolerated at least for short-term therapy. More recently, the HFIM has been utilised for further research into NTM infections. The current therapy for pulmonary *M. avium* complex (MAC) is a macrolide backbone (clarithromycin or azithromycin) along with a companion drug, ethambutol in this case, to which a rifamycin (rifampicin or rifabutin) may be added [40]. However, HFIM experiments with MAC infected monocytes or macrophages treated with azithromycin and ethambutol with or without rifampicin showed poor kill rates and emergence of acquired resistance within 7 days of treatment [41,42]. Promisingly, ceftazidime/avibactam combination showed higher efficacy than the standard backbone treatment against both extra- and intra-cellular MAC at clinically achievable concentrations [43].

Benzylpenicillin was found to be effective against intra-cellular *M. tuberculosis* in the HFIM while the exposure-versus-resistance relationship remained the same as for standard anti-TB drugs [37]. The drug was found to have better early bactericidal activity (EBA) against extra-cellular bacteria compared to isoniazid, a front-line drug with the highest EBA. It also showed sterilising activity against non-replicating cells. Based on the results of the HFIM they confirmed that T_>MIC_ was the best predictor of bacterial kill and recommended dosing regimens. 

Though tetracyclines are not used for the treatment of TB, the positive features of minocycline such as antibacterial potency, high bioavailability, long half-life, good distribution in lung parenchyma, high tolerance, and anti-inflammatory properties justify pre-clinical investigation into its potential as an anti-TB drug. It was found to reduce bacterial loads of both drug-susceptible and resistant strains of *M. tuberculosis*, most likely through apoptotic effects on host cells harbouring the bacteria and could thereby be used for active as well as latent TB [38]. Minocycline was found to be more effective in reducing MAC burden compared to standard treatment at exposures that could be achieved in a majority of patients [44].

Tigecycline, a minocycline derivative, also effectively clears extra- and intra-cellular *M. tuberculosis* as evidenced through HFIM experiments [45]. It mirrored early bactericidal activity of isoniazid while suppressing regrowth of bacteria throughout the course of the experiment. Furthermore, a once-a-week dosing regimen was found to be effective against even rapidly growing bacteria indicating that the injectable drug can feasibly be introduced for MDR-/XDR-treatment without adding excessive burden on stretched health service programmes. Tigecycline has been used by Ferro et al. to investigate activity against *M. abscessus* wherein it showed a 1log_10_ reduction in bacterial burden [46]. Its remarkable efficacy in the HFIM makes the case for tigecycline to become a first line agent for *M. abscessus* infections and its activity against other NTM infections should also be investigated. 

Using HFIM two groups demonstrated that, contrary to expectations, amikacin and moxifloxacin are efficacious against semidormant bacilli at acidic pH [25,47]. In the case of amikacin, the C_MAX_-to-MIC ratio is the best predictor for *M. tuberculosis* killing and this index can be used for individualising therapy where resources allow for it.

## 4. Application of HFIM in Dose and Dosing Interval Selection

Ideally, an optimised dose and dosing interval would obtain maximal bacterial killing whilst suppressing the emergence of resistance and lowering toxicity. However, it is usually a trade-off between these three key criteria.

Linezolid is an oxazolidinone with potent activity against *M. tuberculosis*. Several phase III clinical trials that include linezolid as a part of the combination therapy have shown remarkable success in treating patients with extensively drug-resistant TB [9,48]. However, several toxicities such as gastrointestinal side effects, optic and peripheral neuropathies, and most commonly, haematological toxicity have been reported. Brown et al. used the HFIM to study toxicity, bacterial kill, and emergence of resistance using a range of doses and dosing intervals of linezolid and advised doses ranging between 900 and 600 mg/day that can be reduced to 300 mg/day when exposure-related toxicities are detected [49]. Srivastava et al. also recommended a daily dose of 300 mg/day or an intermittent dose of 1200 mg of linezolid twice a week [50]. Lower doses of linezolid in drug combinations have been tested as part of ZeNix and TB-PRACTECAL trials reporting treatment success [48,51].

Using the clearance rates observed in infants in the HFIM, Deshpande et al. identified the linezolid AUC_0–24_/MIC target for optimal efficacy against TB in children, and an AUC_0–24_ threshold associated with mitochondrial toxicity offering a therapeutic window for treatment [52].

Studies looking into tedizolid as a replacement for linezolid for both child- and adult-TB have reported success in terms of lower toxicity and higher sterilisation rates [53,54]. An RNA-seq approach coupled with HFIM using *M. tuberculosis* H37Ra-infected THP-1 cells confirmed that the treatment produced no signatures indicative of mitochondrial inhibition. Deshpande et al. used the HFIM to investigate the possibility of using linezolid, currently incorporated in treatment of drug resistant TB, to treat pulmonary MAC infections [55]. The result of the study showed that linezolid achieved at least a 1 log_10_ CFU/mL reduction in bacterial burden within a few days. However, the study also showed that linezolid doses which optimise bactericidal effect would be associated with high adverse event rates. Meanwhile, tedizolid was found to be highly bactericidal, even in monotherapy, against MAC [56].

d-cycloserine was first administered for treatment of TB in a clinical setting in 1955. Now, it is mainly used or the treatment of MDR-TB. However, it is neurotoxic and is known to cause several adverse effects such as psychosis and seizures. A systematic review by Deshpande et al. revealed that no studies have been carried out on d-cycloserine that relate AUC with treatment outcomes (cure or relapse) [57]. Thus, they performed a combined exposure-effect and dose fractionation study using the HFIM to determine the PK/PD of the drug. Doses to achieve peak concentrations of 180, 219, and 257 mg/L resulted in higher reduction in bacterial burden in extracellular settings as compared to weekly dosing, as determined by both TTP and CFU analyses. The kill rates obtained were similar to that of first-line drugs and fluoroquinolones. However, owing to its poor penetration in lung cavities as well as reduced efficacy in an intracellular setting, doses between 500 and 750 mg twice daily was recommended by Deshpande et al. for inclusion into the combination therapy used to treat pulmonary TB as well as TB meningitis [57]. It should be noted, though, that HFIM experiments in this study were not performed with the drug in combination with other first- or second-line drugs and the results obtained from these may well vary. 

Ethionamide is a nicotinic acid derivative and a prodrug like isoniazid. It is used to treat MDR-TB. PK/PD studies in HFIM by Deshpande et al. demonstrated that it has a good bacterial kill rate (comparable to isoniazid and ethambutol) [58]. Repeated sampling of the treated bacterial cultures from the hollow fibre cartridge allowed the study of emergence of resistance in these experiments. It was noted that the initial response to treatment was the induction of efflux pumps followed by acquiring mutations in *ethA* and *embA* genes [58]. Therefore, treatment with ethionamide results in phenotypic resistance to drugs such as isoniazid and ethambutol initially through an activated efflux mechanism followed by directed chromosomal mutations in relevant genes.

Fluoroquinolones are a family of drugs that inhibit the enzymes DNA gyrase and topoisomerase IV. Gatifloxacin is a fourth-generation fluoroquinolone and HFIM experiments demonstrated that it cannot achieve complete sterilisation even at exposures of AUC_0–24_/MIC of 678 [59]. Lower exposures resulted in the emergence of resistance within 10–14 days with isolates bearing mutations in the quinolone resistance-determining region of *gyrA* [59]. As in the case of gatifloxacin, the PK/PD of levofloxacin, another fluoroquinoline, is not well examined. Using HFIM along with clinical data and Monte Carlo simulations, Deshpande et al. reported that 800 mg/day dose of levofloxacin was as efficacious in the treatment of pulmonary TB as moxifloxacin [60]. As with gatifloxacin, resistance to levofloxacin emerges by acquiring mutations in *gyrA*; however, only 5% of the population was found to be resistance as compared to 100% as in the case with moxifloxacin or gatifloxacin treatment [60].

Delamanid, a nitroimidazooxazole compound that targets cell wall synthesis in mycobacteria, has been approved for use for MDR-TB (when other treatment options are ineffective) at a dose of 100 mg twice daily. HFIM experiments identified AUC_0–24_/MIC as the index of delamanid efficacy and that a 200 mg daily dose achieved the same response as the twice daily 100 mg dose [61].

## 5. Application of HFIM in Evaluating Combination Therapy

Encouraging results from pre-clinical studies with fluoroquinolones led to trials with regimens containing these drugs aimed at shortening treatment times. However, none of these regimens were found to be non-inferior to the existing standard therapy [62,63]. The HFIM’s strength is in testing the effects of combination therapy as it can mimic human PK profiles of different drugs accurately. HFIM experiments on the PaMZ drug regimen in the STAND trial [8] revealed that the time-to-sterilisation for the above was identical to that of standard therapy, thereby indicating that the regimen being investigated would likely not shorten treatment duration [64]. HFIM experiments in combination with modelling tools can be used to predict the duration of treatment required for a particular combination therapy. Srivastava et al. hypothesise that to shorten treatment time to 2 months, the trajectory of bacterial sterilisation would need to be three times steeper than that of standard therapy [64]. This could serve as a guide for future trials investigating the potential of drug combinations in reduction in treatment time.

Using the HFIM model wherein drugs were tested against *M. tuberculosis* infected mammalian cells, Deshpande et al. [65] recommended doses and dosing intervals of a linezolid and moxifloxacin backbone regimen that could treat tuberculosis in children—a significant problem affecting TB control programmes worldwide. Adding an additional antibiotic, faropenem, at a dose that allowed for T_MIC_ > 62% further improved the efficacy of the treatment and expanded options for treating drug-susceptible and -resistant TB in children as well as TB meningitis [32]. However, adding faropenem (600 mg/daily or twice daily) to a pyrazinamide (40 mg/kg/daily) and linezolid (600 mg/twice daily) combination failed to kill *M. tuberculosis* H37Ra cells faster than standard therapy [66]. This was also found in a tedizolid, faropenem, and moxifloxacin combination study [67]. Elimination of drug combinations that are not superior to standard therapy in sterilisation could be crucial in saving resources and time.

Linezolid and moxifloxacin combinations can be useful in treating MDR-TB cases in adults as well. Using a free bacilli HFIM, Heinrichs et al. demonstrated that 900 mg/day of linezolid was efficacious even in acidified environments, and a dose of 600 mg/day as a part of a robust combination could eliminate bacteria and suppress the rise of resistance [68]. Contrary to reports by Louie et al., they found moxifloxacin to lose efficacy as the pH of the medium was lowered, and recommended a dose of 800 mg/day [25,68]. Moxifloxacin in combination with thioridazine, a repurposed antipsychotic drug, exhibits significant bactericidal activity against intracellular MAC with bacterial burdens dropping below the limit of detection (0.3 log_10_ CFU/mL) [69].

Srivastava et al. reported that drug combinations of fluoroquinolones (800 mg of moxifloxacin) with increased doses of rifampicin (3 times the standard dose) and pyrazinamide (twice the standard dose) result in rapid killing of *M. tuberculosis* H37Ra with no significant toxicity as demonstrated by the liver organoid model developed in the hollow fibre system [70]. Interestingly, higher doses of rifampicin administered as monotherapy did not yield higher antimycobacterial activity in HFIM [71].

Combinations of vancomycin with d-cycloserine or benzylpenicillin, all cell wall-targeting drugs revealed synergistic and antagonistic relationships, respectively [72].

The HFIM and its adaptations can identify drug combinations with efficacy at par or better than standard therapy, indicate length of treatment for sterilisation, reveal drug–drug interactions, and offer insight into toxicity making it a powerful tool that should be more widely used at the pre-clinical stages for anti-TB drug development.

## 6. Discussion

As with all models, the HFIM presents some challenges and disadvantages. The in vitro nature of the model does not allow investigation of the dynamics of antimicrobial treatment in the presence of a functional immune system. However, the absence of immune cells allows us to attribute the observed changes directly to the antimicrobial regimen administered, thus providing important pre-clinical data. Whilst using HFIM, we cannot model antimicrobial penetration to the infection site; however, pharmacokinetic data from human samples defining the concentrations reached at sites of infection can be mimicked in the HFIM if available. Pro-drugs that require a specific environment for activation (for example, pyrazinamide) cannot be used in HFIM without replicating the host physiology [73]. Other drugs, such as bedaquiline, undergo metabolism by the cytochrome enzymes to yield metabolites with similar or lower potency [74]. Again, if the proportion of the metabolites is known, and they are available commercially, concentrations observed in patient can be mimicked in the HFIM. Plasma protein binding of administered drugs is an important factor that contributes to both the pharmacokinetics and pharmacodynamic characteristics. Current HFIM models do not mimic the plasma protein binding of antimicrobials. This is factored in while calculating the doses that are administered. Therefore, adapting the model to integrate protein binding is an important methodological advancement that should be considered. Currently, the mouse thigh infection model remains the gold standard for antimicrobial PK/PD; however, with increased use of the HFIM and attention to standardization of protocols and outcome measures [14], HFIM has a significant role to play.

Treatment of mycobacterial infections requires a lengthy combination therapy for successful outcomes and this complicates protocols, both in vitro and in vivo, for drug development. As we entered the 21st Century, there was substantial investment in three phase 2c/3 trials seeking treatment shortening through the switching of a fluoroquinolone into the standard TB treatment regimen, largely based on mouse model data; none of these studies resulted in a treatment shortening protocol [62,63,75]. Given the investment of time and money, there is a need to reduce the risk of these endeavours by improving the information obtained in pre-clinical stages [76]. The HFIM allows investigation of the early bactericidal effects, consequently providing information on the intensive phase of treatment when bacillary load is still observed in sputum [77]. As the host immune response, apart from investigations involving *M. tuberculosis* infection models in monocytes or macrophages, cannot be followed, the HFIM should therefore be seen as an addition to pre-clinical animal experiments to characterise and prioritise drug combinations prior to going into human studies. This allows evaluation of the hypothesis in a relatively inexpensive way prior to Phase II studies in human. Advancements in in vitro PK/PD models, such as the HFIM, need to be supported by development in assays to measure drug penetration at the site of infection and its efficacy under the widely differing physiological environments created during infection. Developments in organ-on-a-chip platforms and three-dimensional in vitro granuloma models offer promise in this regard [78,79]. Integrating information from all of these platforms is key for translational success. The good news is that the anti-mycobacterial drug development pathway is opening up [80]. In order to benefit from the new compounds, we need tools to drive evaluation, allowing us to prioritise individual antibiotics and novel combinations; the HFIM has an important role in this process.

## Figures and Tables

**Figure 1 antibiotics-10-01515-f001:**
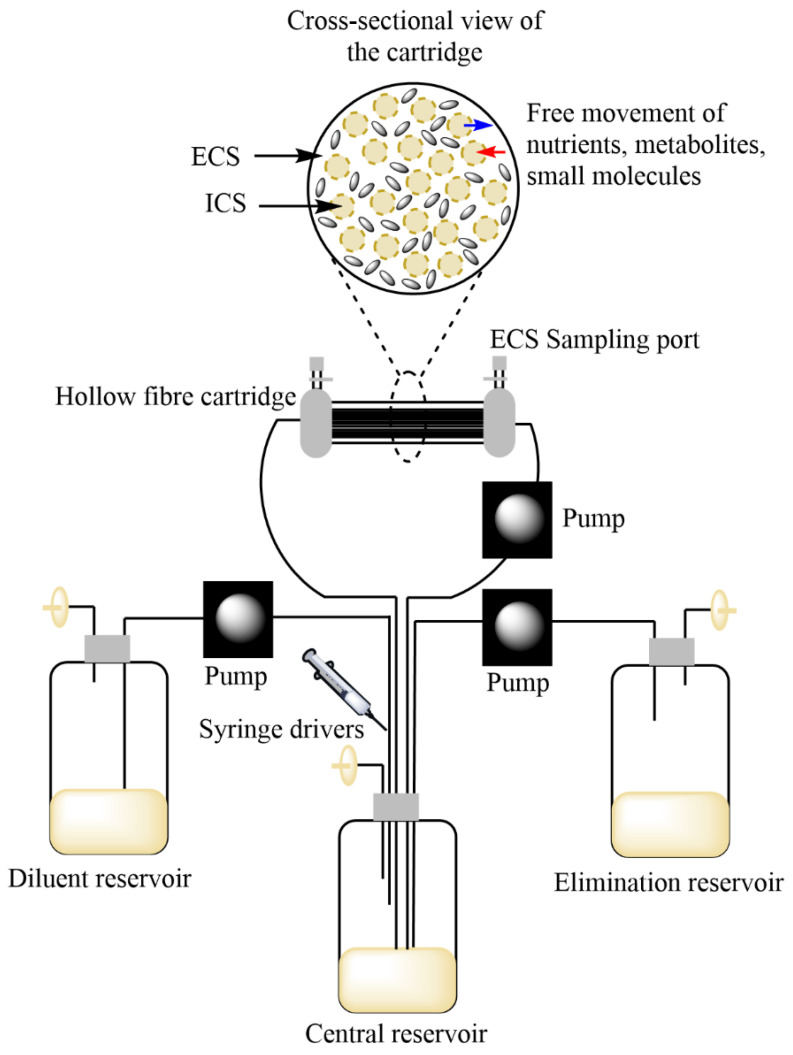
The HFIM setup: Culture medium is withdrawn from the diluent reservoir into the central reservoir, which also receives drugs via computerised syringe drivers. The media circulates between the hollow fibre cartridge and the central reservoir and is extracted to the waste reservoir at rates pre-defined by the elimination profiles of the drugs being investigated. While small molecules such as drugs and nutrient pass freely across the pores of the fibres between the intra-and extra-capillary space (ICS/ECS), the bacteria are contained within the ECS and can be withdrawn from the sampling ports for downstream analyses.

**Figure 2 antibiotics-10-01515-f002:**
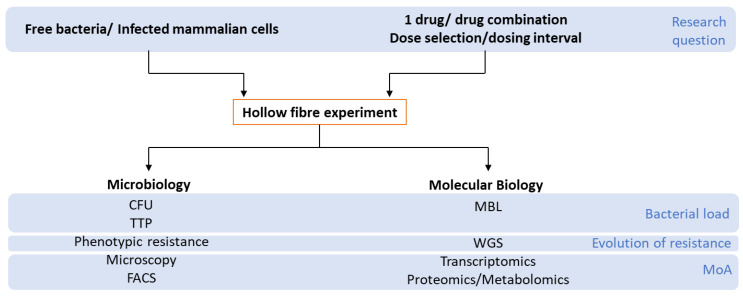
The HFIM working model: This technique can be adapted to most extra-/intra-cellular pathogens under single or combination drug therapy. Drug concentration profiles and media components can be altered to mimic the host infection sites and a multitude of readouts can be obtained from serial sampling of the bacteria to investigate the effects of treatment and its mechanism of action (MoA). CFU—colony forming units; TTP—time to positivity; FACS—cell sorting; MBL—Mycobacterium bacterial load; WGS—whole-genome sequencing.

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
