# Peer review of "Improving the Drug Development Pipeline for Mycobacteria: Modelling Antibiotic Exposure in the Hollow Fibre Infection Model"

_antibiotics, 2021, doi:10.3390/antibiotics10121515_

Round 1
Reviewer 1 Report
Regarding the paper entitled: “Improving the drug development pipeline for mycobacteria: modelling antibiotic exposure in the hollow fibre infection model”, the manuscript was well written and provides a historical perspective and highlights recent advancements/challenges pertaining to antibiotic modelling in the hollow fibre infection model. I recommend this article to be accepted for publication in Antiobiotics provided the following points are addressed.
Line 180: 12 mg in adults needs to be changed to 12 g in adults
Line 361 2c/3 trails should be trials
Line 424 re: reference 14: It appears that 8 authors were left out from this citation. Please include these 8 authors.
Author Response
The authors thank the reviewer for their comments and have made the changes suggested.
Line 180: 12 mg in adults needs to be changed to 12 g in adults.
--Thank you for bringing this error to our notice. It has been corrected.
Line 361 2c/3 trails should be trials.
--Done.
Line 424 re: reference 14: It appears that 8 authors were left out from this citation. Please include these 8 authors.
--The reviewer correctly pointed out that the citation was not complete and has been completed as follows: “Sadouki, Z.; McHugh, T.D.; Aarnoutse, R.; Ortiz Canseco, J.; Darlow, C.; Hope, W.; van Ingen, J.; Longshaw, C.; Manissero, D.; Mead, A.; Pelligand, L.; Phee, L.; Readman, J.; Ruth, M.M.; Standing, J.F.; Stone, N.; Wey, E.Q.; Kloprogge, F. Application of the hollow fibre infection model (HFIM) in antimicrobial development: a systematic review and recommendations of reporting. J. Antimicrob. Chemother. 2021, 76, 2252–2259.“
Reviewer 2 Report
The authors reviewed the hollow fiber infection model in the preclinical treatment of mycobacterial infection. The advantages and disadvantages were discussed. Generally, the paper is well organized and fits the scope of the special issue. A minor revision is recommended before publication:
1) The disadvantages including lacking tissue penetration and the possible metabolic degradation could be discussed.
2) The spelling should be checked.
Author Response
The authors thank the reviewer for their constructive comments.
1) The disadvantages including lacking tissue penetration and the possible metabolic degradation could be discussed.
--The reviewer has raised valid points. When pharmacokinetic data from human samples defining the concentrations reached at sites of infection is available, they can be mimicked on the HFIM. This is discussed in lines 347-350. We have discussed drug metabolism as below:
“Whilst using HFIM we cannot model antimicrobial penetration to the infection site, pharmacokinetic data from human samples defining the concentrations reached at sites of infection can be mimicked in the HFIM if available. Pro-drugs that require a specific environment for activation (for example, pyrazinamide) cannot be used in HFIM without replicating the host physiology [73]. Other drugs, such as bedaquiline, undergo metabolism by the cytochrome enzymes to yield metabolites with similar or lower potency [74]. Again, if the proportion of the metabolites is known, and they are available commercially, concentrations observed in patient can be mimicked in the HFIM.”
2) The spelling should be checked.
--The manuscript has been thoroughly proofread before resubmission.
Reviewer 3 Report
Dear authors, this is a nice review of an important topic. All the salient points are there, and explained correctly. It is well written, flows well, and I have no issues recommending this for publication.
Minor comments: line 360 to the end: this sounds like more personal discourse rather than scientific literature. Perhaps the authors can add some references, and for example, it would be good to have examples for line 382
Author Response
The authors thank the reviewer for their constructive suggestions.
Minor comments: line 360 to the end: this sounds like more personal discourse rather than scientific literature. Perhaps the authors can add some references, and for example, it would be good to have examples for line 382
- The authors appreciate the reviewer’s comments. We have added examples to line 382, references where necessary and removed extraneous sentences. Altered text now reads as below:
- “As we entered the 21st Century there was substantial investment in 3 phase 2c/3 trials seeking treatment shortening through the switching of a fluoroquinolone into the standard TB treatment regimen, largely based on mouse model data; none of these studies resulted in a treatment shortening protocol [62,63,75]. Given the investment of time and money, there is a need to reduce the risk of these endeavours by improving the information obtained in pre-clinical stages [76]. The HFIM allows investigation of the early bactericidal effects; consequently provides information on the intensive phase of treatment when bacillary load is still observed in sputum [77]. As the host immune response, apart from investigations involving tuberculosis infection models in monocytes or macrophages cannot be followed, the HFIM should therefore be seen as an addition to pre-clinical animal experiments to characterise and prioritise drug combinations prior to going into human studies. This allows evaluation of the hypothesis in a relatively inexpensive way prior to Phase II studies in human. Advancements in in vitro PK/PD models, such as the HFIM, need to be supported by development in assays to measure drug penetration at the site of infection and its efficacy under the widely differing physiological environments created during infection. Developments in organ-on-a-chip platforms and three-dimensional in vitro granuloma models offer promise in this regard [78, 79]. Integrating information from all of these platforms is key for translational success. The good news is that the anti-mycobacterial drug development pathway is opening up [80].”